# Groundwater and baseflow drought responses to synthetic recharge stress tests

Jost Hellwig[1], Michael Stoelzle[1], Kerstin Stahl[1,2]

[1]Environmental Hydrological Systems, University of Freiburg, Freiburg, 79085, Germany
[2]Freiburg Institute of Advanced Studies (FRIAS), University of Freiburg, Germany

*Correspondence to*: Jost Hellwig (jost.hellwig@hydrology.uni-freiburg.de)

**Abstract.** Groundwater is the main source of freshwater and maintains streamflow during drought. Potential future groundwater and baseflow drought hazards depend on the systems' sensitivity to altered recharge conditions. We performed
groundwater model experiments using three different generic stress tests to estimate the groundwater- and baseflow drought sensitivity to changes in recharge. The stress tests stem from a stakeholder co-design process that specifically followed the idea of altering known drought events from the past, i.e. asking whether altered recharge could have made a particular event worse. Across Germany groundwater responses to the stress tests are highly heterogeneous with groundwater heads in the North more sensitive to long-term recharge and in the Central German Uplands to short-term recharge variations. Baseflow
droughts are generally more sensitive to intra-annual dynamics and baseflow responses to the stress tests are smaller compared to the groundwater heads. The groundwater drought recovery time is mainly driven by the hydrogeological conditions with slow (fast) recovery in the porous (fractured rock) aquifers. In general, a seasonal shift of recharge (i.e. less summer recharge and more winter recharge) will therefore have low effects on groundwater and baseflow drought severity. A lengthening of dry spells might cause much stronger responses, especially in regions with slow groundwater response to precipitation. Water
management may need to consider the spatially different sensitivities of the groundwater system and the potential for more severe groundwater droughts in the large porous aquifers following prolonged meteorological droughts, particularly in the context of climate change projections indicating stronger seasonality and more severe drought events.

## 1 Introduction

Freshwater is a vital resource for human life and the demand is growing worldwide simultaneously to economic and
demographic growth. The largest accessible storage and one of the most important sources for human water demand is groundwater (Gleeson et al., 2016; Wada et al., 2014), especially in case of low surface water availability, and it is expected to become even more important under climate change (Taylor et al., 2013; Kundzewicz and Döll, 2009). Groundwater serves as a buffer against hydroclimatic variations and is a considerable factor influencing the propagation of drought (Eltahir and Yeh, 1999; Peters et al., 2003). Drought is defined as below normal water availability and starts with a meteorological drought
that can propagate through all parts of the hydrological cycle (Van Loon, 2015). It can lead to social and economic impacts,

especially during seasons with low water availability compared to water demand. As a natural hazard drought affects people worldwide and causes high economic loss (EC, 2007). Hence, the groundwater's potential to attenuate meteorological droughts influences society´s current and future vulnerability to drought events.

The groundwater response to meteorology can be highly diverse both on small and large scales (Stoelzle et al., 2014; Bloomfield et al. 2015; Kumar et al., 2016, Haas and Birk, 2018). Weider and Boutt (2010) showed that groundwater responses to precipitation anomalies are more heterogeneous compared to the responses of streamflow. Accordingly, Bloomfield et al. (2015), Kumar et al. (2016) and Stoelzle et al. (2014) consistently found that typical time scales of drought propagation into groundwater are site-specific, pointing to the importance of hydrogeological characteristics and subsurface storage processes. The sensitivity to changes in the meteorology will hence be site-specific and is often not generalizable, in particular when considering borehole data from specific locations within an aquifer and relative to rivers or recharge areas (Heudorfer and Stahl, 2017). Hellwig and Stahl (2018) found that the differences in the groundwater response to precipitation anomalies also correspond to varying sensitivities of baseflow to precipitation shifts.

To assess the groundwater and baseflow sensitivity to changes in climatic conditions on larger scales, extensive observational data capturing the large diversity of their responses to meteorology would be required. However, unlike surface water, groundwater is hard to observe on larger scales in sufficient resolution for these analyses. As borehole observations are often hardly scalable (Kumar et al., 2016) they are usually not sufficient to investigate groundwater sensitivity to climate variability on larger scales. Therefore, groundwater models are often inevitable for detailed investigations. Recently, the use of large-scale groundwater models including gradient driven lateral flows has gained increasing attention (e.g. Maxwell et al., 2015; de Graaf et al., 2015; Reinecke et al., 2019), as large-scale datasets on aquifer parameters become increasingly available. Hellwig et al. (2020) demonstrated that these models can depict the differences in propagation time from meteorological water deficits to groundwater (droughts) on larger scales reasonably well, concluding that they are also suitable to assess the groundwater's and baseflow's sensitivity to recharge changes on larger scales.

A systematic assessment of sensitivities is often realised based on a scenario-neutral ensemble approach, for example, to inform planning processes for floods (Prudhomme et al., 2010). Other than commonly used scenarios based on climate change projections, scenario-neutral approaches aim to provide robust information on potential change directions based on the system's characteristics and independent from specific emission scenarios and climate change uncertainties. Unlike climate change scenarios which provide probabilities of changes based on specific projection assumptions, scenario-neutral stress tests explore the systems' general responsiveness, e.g. to other environmental changes or to extreme events. Therefore, stress tests must not be interpreted as predictions of future conditions but rather provide information on system sensitivities for management or adaptation planning. Designing stress tests for drought, a slowly developing phenomenon with time lagged signal in streamflow and groundwater, requires the consideration of long lead times and resulting depletion of catchment storage. For example, Staudinger et al. (2015) used model experiments of progressive drying to assess the streamflow sensitivity to drought for catchments across Switzerland. Stoelzle et al. (2014) developed a model-based stress test approach to study the sensitivity of streamflow to changes in climate based on modifications of the recharge. More applied synthetic

stress-testing approaches often use worst case scenarios to estimate the consequences of specific events (Stoelzle et al., 2020b). Stress-testing sensitivity to drought will help to better understand the degree of resilience of various hydrological systems (Hall and Leng, 2019).

As part of the Climate and Water Initiative of southern Germany's federal states (KLIWA) different types of stress tests or "what-if experiments" were explored as means to better understand and more easily communicate potential future changes to
low flow (Stoelzle et al., 2018; 2020b). Stress test designing included for example a progressive recharge reduction before the 2003 summer drought, as this event is often used as planning benchmark or to assess follow-up costs: the stress tests ask whether the effect may even have been worse, e.g. with different antecedent conditions. The co-design process of KLIWA revealed different preferences, including rather arbitrary repetitions of sequences of past (known) dry years, very straightforward 'wetter-drier' modifications of past periods or specific drought events, and more systematic approaches with
larger model ensembles of modified conditions. In this study we employ three of the approaches from this co-design process that also allow for a systematic analysis of stress responses (e.g. drought recovery).

Specifically, the stress tests focus on pre-drought recharge reduction effects on the hydrological drought sensitivity simulated in the groundwater-baseflow domain. Directly modifying groundwater recharge allows to focus the research question to the storage-outflow processes relevant to the hydrology in dry periods. In this study this modification aims at testing and attributing
specific system sensitivities rather than an overall system response to climatic change projections. As groundwater has a recharge memory, antecedent recharge conditions are a key factor for groundwater drought severity and the effect of perturbed recharge on drought severity can provide information on the site-specific groundwater and baseflow drought sensitivity. The approach by Stoelzle et al. (2014) illustrated an assessment of the sensitivity to altered recharge in reservoir or box-type hydrological models and was limited to the investigation of baseflow sensitivity.

In this study, we use similar recharge stress tests, as well as the ideas of KLIWA, for entire Germany in a large-scale high-resolution MODFLOW-groundwater model to assess a range of potential changes to groundwater and baseflow drought hazard. Specifically, this study aims to

(1)      assess the sensitivity of groundwater and baseflow drought to a seasonal wetting and drying shift,

(2)      identify large-scale sensitivity patterns of groundwater and baseflow drought events to extreme recharge drought
conditions with particular return periods, and

(3)      quantify characteristic groundwater drought recovery times.

## 2 Study area and groundwater model setup

The study area of this work is the state of Germany. Germany consists of four major geographical regions with different groundwater characteristics (Figure 1): the lowlands in the North with slow responding groundwater in porous aquifers, the
uplands in Central Germany with faster responses and mixed aquifer types including fractured rocks and karst aquifers, the Alpine foothills in southern Germany with porous aquifers and the high elevation Alps in the far South with mostly fractured

rocks aquifers. Germany's temperate humid climate is characterized by evenly distributed precipitation throughout the year and an annual temperature cycle that results in climatic water deficits due to higher evapotranspiration rates. As a result, groundwater recharge largely takes place during the winter months (Jacob et al., 2012; Kopp et al., 2018). Future climate

projections indicate – despite all uncertainties emerging from different models and emission pathways – as a general pattern that precipitation will increase during winter (between -10% and + 20%) and decrease during summer (between -30% and + 10%) (e.g. Jacob et al., 2012; Paparrizos et al., 2018; Herrmann et al., 2016). Combined with increasing temperatures over the whole year, recharge will most likely increase in winter and decrease in summer (Eckhardt and Ulbrich, 2003; Stoll et al., 2011; Dams et al., 2012; Hunkeler et al., 2014; Chen et al., 2018). The magnitude of change is highly uncertain with low model

agreement compared to other regions in the world (e.g. Reinecke et al., 2019) and depends on the choice of recharge model (e.g. Moeck et al., 2016) as well as the choice of compared reference and future periods.

To assess the groundwater response to recharge stress tests we applied a large-scale groundwater model covering Germany. The model consists of one MODFLOW layer (Harbaugh et al., 2000), simulating groundwater heads, baseflow (i.e. groundwater discharge to surface water) and lateral flows in weekly time steps. It covers all basins intersecting Germany (i.e.

river Rhine in the West, river Danube in the South, river Elbe and river Oder in the East) with a spatial resolution of approximately 1 km (latitudinal: 1/22°, longitudinal: 1/14°). Hellwig et al. (2020) developed and evaluated the model, demonstrating its ability to depict the heterogeneous groundwater response to precipitation anomalies even though model performance markedly declined in the mountainous regions due to the larger topographic variability. In the following the model structure and input data are briefly described, for detailed information refer to Hellwig et al. (2020).

Specific yield values were taken from the porosity values in the GLobal HYdrogeology MaPS (GLHYMPS: Gleeson et al., 2014). Initial hydraulic conductivity values $k_0$ for Germany were derived from the "Hydrogeologische Übersichtskarte" (hydrogeological map HÜK200: BGR and SGD, 2016), for the rest of the model domain $k_0$ was based on GLHYMPS' permeability values. Consistent with other groundwater models based on a single layer (e.g. Fan et al., 2007; Miguez-Macho et al., 2008), hydraulic conductivity was assumed to decrease exponentially with depth. The characteristic decrease is described

by an exponential spatially varying depth function $f$ which inversely relates hydraulic conductivity to the slope of surface terrain (i.e. a faster decrease of conductivity with depth in areas with steeper slopes). Then, transmissivity $T$ depends on $k_0$, $f$ and the current groundwater table depth $d_{gw}$:

$$T = \int_{d_{gw}}^{100} k_0 e^{\frac{-z'}{f}} \, dz' \tag{1}$$

where $z'$ is the depth below surface and $T$ is updated every time step.

Interactions between surface water and groundwater were implemented using the RIV-package, simulating flow dependent on the difference of groundwater and surface water heads. Each cell contains either a large river (width > 10 m) with strong interactions with the aquifer or a small stream (width < 10 m) with less interactions. Channel depth, riverbed conductivities and river head over riverbed were derived from long-term average routed baseflow of previous model runs (Hellwig et al.,

2020). Baseflow and infiltration was assumed to be proportional to the difference of groundwater heads and surface water heads as well as riverbed conductivity. Hence, with decreasing water tables baseflow reduces and stops when groundwater heads fall below surface water heads.

Groundwater recharge was calculated using a conceptual recharge model consisting of a soil storage and a snow storage. Rainfall, snow and evaporation (following Hargreaves and Samani, 1985) were derived from the European Climate Assessment & Dataset (Haylock et al., 2008), version 16. The soil storage was parameterized with data from the 'Hydrologischer Atlas Deutschlands' (HAD, hydrological atlas of Germany; BMU, 2003). To ensure realistic recharge rates, recharge was rescaled using long-term average recharge estimates from the HAD.

This study uses time series of water table and baseflow dynamics from 1970 to 2016 (reference run). For different stress tests, recharge and boundary conditions in the model are altered and resulting water table and baseflow time series are compared to the reference run.

## 3 Stress test design and modelling approach

Three types of generic recharge stress tests addressing different questions for drought management were applied to the groundwater model (Table 1). To do this the stress tests have different boundary conditions and different recharge modifications. All stress tests apply relative changes over entire Germany, thus allowing the results to be analysed as composite maps of the same relative change but with respect to the specific local conditions. This sensitivity analysis approach should not be confused with the more common climate change model chain experiments that would apply locally varying changes stemming from the combination of climate model output and hydrology or soil water balance models with particular assumptions and parametrizations of vegetation and soils. The composite maps therefore represent response differences to the designed stress test inputs due to hydrogeology.

The first stress test $S_{SHIFT}$ assumes a change in drought hazard due to an increased seasonality of precipitation and temperature. This stress test aims to answer practitioners' questions how an intra-annual climatic shift in Germany can affect inter-annual variability as well as extreme events such as droughts in groundwater and baseflow (Table 1). Therefore, for $S_{SHIFT}$ precipitation is assumed to increase in winter and decrease in summer whereas temperature increases over the whole year. The stress test experiment consequently increases (decreases) recharge during winter (summer), directly amplifying recharge seasonality. The model is run from 1970 to 2016 with different assumptions of the magnitude (5, 10, 15, 20, 30%) of recharge shift from a decrease during summer months (JJA) to an increase during winter months (DJF) (Figure 2).

For the assessment of the response to $S_{SHIFT}$ we compare the variability for different seasons (i.e. variability is calculated for water table/baseflow of selected months taken from all simulated years) and percentile thresholds for water table/baseflow during drought from the stress test run with the reference run forced by original recharge. As a spatially and temporally varying threshold $\tau$ we use 0.10, 0.25 and 0.50 representing an exceedance probability of 90, 75 and 50% within the specific season

(Van Loon and Van Lanen, 2012; Heudorfer and Stahl, 2017). An increase (decrease) of the water table/baseflow under $S_{SHIFT}$ indicates a higher (lower) water availability for the selected drought severity.

The second stress test type $S_{EVENT}$ focuses directly on the scale of selected drought events and is designed to assess the groundwater's drought sensitivity to systematic changes in the antecedent recharge conditions (Table 1). Practitioners often

use past events for the design of drought management plans and ask whether there might be conditions that had the potential to make similar events even worse (Table 1). For this study the events of 1973, 2003 and 2015 are selected for the analysis of a range of different but well-known severe benchmark drought years. These drought years have received attention in previous publications, and although they all had large precipitation deficits also differences were noted (e.g. Tallaksen and Stahl, 2014; Laaha et al., 2017; Hellwig, 2019; Hellwig et al., 2020). Due to differences in the recharge conditions before the droughts, the

groundwater situation was very different in each case (Hellwig, 2019). While the 1973 event can be characterized as a long-term water deficit leading to depleted water tables across Germany (Figure 3a), the events in 2003 and 2015 were rather severe short-term summer drought events. As the winter 2002/03 was exceptionally wet, most water tables were not depleted in summer 2003 (Figure 3b). The 2015 event followed a winter of average recharge and led to a severe groundwater drought in the following summer in the fast responding aquifers in the South whereas the slower responding aquifers in the North did not

develop anomalies corresponding to a groundwater drought (Figure 3c). With $S_{EVENT}$ these real antecedent recharge conditions for every modelled grid cell were further stressed by altering recharge for three different durations (3, 9 and 24 months) to investigate different time scales. The month of the groundwater drought's start is set in May. For the 3-month (9-month, 24-month) stress tests we modify recharge backwards from the drought's start for 3 (9, 24) months starting in February (August of the year before, May two years before) and compare the resulting groundwater situation from May to November in the

drought year to the reference simulation (Figure 2).

The amount of antecedent recharge is modified to represent a "recharge deficit event" with a return period $T_{RP}$ of 50 and 100 years based on the modelled 57 years of reference recharge series for each grid cell (1960-2016). The use of return periods allows a consistent spatial comparison of the same stress test intensity. First, for all three durations the corresponding 57 recharge sums are used to fit a generalized extreme value distribution with Weibull plotting positions. Then, fitted distributions

are used to estimate the recharge sums of drought events with $T_{RP} = 50$ and $T_{RP} = 100$ years representing different drought severities. Finally, the reference recharge time series is rescaled to match these recharge sums while conserving the original variability of the recharge time series (Stoelzle et al., 2014). The reduced recharge is then used as an input for the groundwater model. Altogether, this stress test type consists of 18 model runs: for 3 drought years (1973, 2003, 2015) antecedent recharge is modified on three time-scales (3, 9, 24 months) to match that of a drought event of two return periods (50y, 100y).

For the assessment of the response to $S_{EVENT}$ we analyse changes in water table/baseflow for all different benchmark droughts, time scales and return periods. Effects of $S_{EVENT}$ are related to potential explanatory variables from the groundwater model: hydraulic conductivity, specific yield, elevation, slope, aquifer type and precipitation accumulation times that have the

maximum correlation ($T_{max}$) with groundwater and baseflow. $T_{max}$ can be understood as the time scale of anomaly propagation from climate to the groundwater system and ranges between one month and several years.


The third stress test type ($S_{RECOV}$) is strictly speaking not a test that applies additional stress but a test of system recovery. It focuses on the recovery of the worst drought events in the historical record and aims to answer practitioners' question how long the drought will last if the following months are normal, dry or wet (Table 1). As groundwater dynamics are often more damped than climate anomalies, groundwater droughts usually last longer than meteorological droughts. To assess the
maximum duration the groundwater system needs to recover from severe drought conditions, the lowest groundwater heads simulated between 1970 and 2016 are taken as the initial condition for each grid cell in this simulation experiment. Then, starting in October (in general, the beginning of the main recharge period in Germany), groundwater heads are simulated using three assumed recharge tests as input: average monthly recharge, continuously dry (25-percentile monthly recharge) and wet (75-percentile monthly recharge) recharge conditions, derived from the long-term historical recharge record (Figure 2).
Drought termination is set to when the simulation exceeds the recovery threshold for the first time. As a recovery threshold we also test three options: the monthly variable 25-percentile groundwater head (i.e. the groundwater head that is exceeded 75% of the time in that calendar month considering all simulated years), and the 40- and 50-percentile groundwater head. The time between each simulation start and the drought termination is the groundwater recovery time $T_{rec}$, i.e. the time needed to recover from the worst drought conditions. Like for the interpretation of the results from $S_{EVENT}$ we relate $T_{rec}$ to potential
explanatory variables.

The groundwater model used for these experiments was evaluated by Hellwig et al. (2020) using 202 groundwater borehole time series and 338 streamflow observations. Their results suggested that the model can reproduce the standardized time series as well as $T_{max}$, even though the model is still too coarse for the small-scale variability in mountainous regions of Germany. However, for the different stress tests, specific model abilities will be required (Table 2). While for $S_{SHIFT}$ the appropriate
simulation of $T_{max}$ measuring the time needed to propagate anomalies from precipitation to groundwater is most relevant, for $S_{EVENT}$ it is more the depiction of drought severity during the selected benchmark drought events. These two model abilities are also essential for $S_{RECOV}$. In general, overall patterns of the stress test results can be expected to be reliable for both groundwater heads and baseflow with largest uncertainties of the actual groundwater levels and the magnitude of their fluctuations in the porous aquifers in North-East and the mountainous South.

**4 Results**

**4.1 Groundwater drought under a seasonal recharge shift**

The assumed $S_{SHIFT}$ affects groundwater heads and baseflow throughout the year. As recharge increases (decreases) during winter (summer) recharge variability increases (decreases) correspondingly (Figure 4). Most recharge in Germany (outside the Alps) occurs during winter, therefore, the seasonal differences are amplified by $S_{SHIFT}$ and inter-annual variability for recharge

as well as groundwater tables and baseflow is increased. While in general, the changes in seasonal baseflow variability correspond to the changes in recharge variability, alterations of groundwater head variability are much more heterogeneous. Not only in winter but also during spring and autumn there is an increase in variability across large parts of Germany and even in summer variability increases in the Northeast.

Under $S_{SHIFT}$ groundwater heads increase due to the higher winter recharge except in the alpine South, where groundwater recharge mostly occurs during summer (Figure 5). Changes of groundwater heads are smaller during drought than for median conditions, with negligible differences between the seasons. Absolute head changes are stronger in aquifers of large head variability (i.e. the fractured rock aquifers). On the contrary, relative head changes standardized by the mean and standard deviation of natural variability are most pronounced in the large porous aquifers in the North (Figure S2) where changes of variability are strongest as well (Figure 4). The general pattern of head changes is similar for all different assumed shift magnitudes (Figure S3).

Baseflow also increases under $S_{SHIFT}$ in most parts of Germany (Figure 6). However, there are differences between the seasons: during winter there is a large increase of baseflow, particularly under average conditions. In spring and autumn there are only small increases in the north of Germany (not shown). Baseflow changes during summer are bidirectional with increases in the North and decreases in the South, again more pronounced for average conditions than for drought. On an annual scale changes in baseflow are rather small following the same pattern of increases in the North and decreases in the South. Changes of baseflow relative to its variability are in general much smaller compared to changes of groundwater heads (Figure S4). As for groundwater patterns of baseflow changes are independent from the assumed shift magnitude with stronger responses for larger relative recharge shifts (Figure S3).

## 4.2 The groundwater drought sensitivity to antecedent recharge

All $S_{EVENT}$ stress tests exacerbate the selected benchmark groundwater droughts (Figure 7). However, the magnitude of declines in groundwater head and baseflow vary for different drought events and durations. In comparison, the effect of the chosen return period is low. The differences between $S_{EVENT}$ with $T_{RP} = 50y$ and $T_{RP} = 100y$ are about one order of magnitude smaller than the differences among the different $T_{RP} = 50y$ recharge reduction durations. The median deviation to the reference simulation ranges between 4 % and 21 % for the different $S_{EVENT}$.

Differences between the drought events are similar for water table and baseflow changes (Figure 7). For the 1973 drought event declines are most pronounced for a reduced recharge over 3-months whereas for the short-term summer droughts in 2003 and 2015 longer durations of recharge reductions cause more severe declines. However, the magnitude of stress test caused decreases is different for water tables and baseflow. Water table declines are largest for stress tests of the 2003 drought and smallest for the 1973 drought (Figure 7a) whereas relative baseflow decreases are similar for all events (Figure 7b). The differences between the stress tests as well as water tables and baseflow also show distinct spatial patterns (Figures S5-S6). For example, for the 3-months duration only specific regions in the Central German Uplands are affected with most pronounced head declines for the 1973 event.

The effects of $S_{EVENT}$ are related to different parameters (examples in Figures S7-S8), most significantly to the anomaly propagation time $T_{max}$. In general, longer $T_{max}$ are related to stronger head decreases whereas baseflow reductions are larger for shorter $T_{max}$ (Figure 8). However, the exact relationship between $T_{max}$ and stress test depends on the event year and duration of the recharge reduction.

## 4.3 Recovery times of groundwater drought

Consistent with the results from $S_{SHIFT}$ and $S_{EVENT}$, there is a large heterogeneity of $T_{rec}$ across Germany (Figure 9). For average recharge conditions and a 25-percentile recovery threshold $T_{rec}$ is shorter than 10 months in large parts of Germany, particularly in the Central German Uplands with its fractured rock aquifers (Figure 9a). In these regions, a single average recharge season can be enough to terminate a severe groundwater drought. In the north-eastern part of Germany, which is characterized by large porous aquifers, groundwater heads will still not recover to the 25-percentile recovery threshold after up to 60 months of average recharge. In these regions, average recharge is not enough to terminate a severe groundwater drought. Accordingly, a bi-modal distribution of $T_{rec}$ is found for regions with fast recovery and for regions with no recovery at all in the timeframe. For dry recharge conditions most of Germany will not recover within 60 months apart from some fast responding regions in the Central German Uplands (Figure 9b). On the contrary there are only few regions (most of them in the northeast of Germany) that do not recover within a year given continuously wet recharge conditions (Figure 9c). The larger recovery thresholds lead to increased $T_{rec}$ but the general spatial pattern of regions with slower and faster recovery remains the same (not shown). $T_{rec}$ increases with hydraulic conductivity and specific yield used in the model grid cell and is significantly higher in porous aquifers compared to aquifers in fractured rocks (Figure 9b). However, the strongest relationship is found between $T_{rec}$ and propagation time $T_{max}$. The strong relationship between $T_{max}$ and $T_{rec}$ is found for all $S_{RECOV}$ independent from the choice of recharge conditions and recovery threshold.

## 5 Discussion

### 5.1 Groundwater and baseflow sensitivity to altered recharge

All stress tests revealed a spatially highly heterogeneous groundwater response due to changes in recharge. In the northeast of Germany where large porous aquifers are prevalent, groundwater heads respond to long-term recharge characteristics. Accordingly, in this region changes on the 24-months duration ($S_{EVENT}$) or changes in the annual average recharge sum ($S_{SHIFT}$) cause the strongest responses. Contrasting, in the fractured aquifers of the Central German Uplands intra-annual recharge dynamics are much more relevant, demonstrated by the stronger responses to 3-months stress tests ($S_{EVENT}$). Also, the recovery time $T_{rec}$ from a severe drought showed the same patterns with faster recovery in the uplands and slower recovery in the large porous aquifers ($S_{RECOV}$). These results highlight the importance of the hydrogeological characteristics for assessing the groundwaters' sensitivity to drought and for drought propagation, supporting the findings of Stoelzle et al. (2014).

Inter- and intra-annual changes in recharge do not only affect the immediate drought hazard in a different way for different hydrogeology but will also cause various changes to the long-term groundwater and baseflow dynamics. A change of recharge

variability will not necessarily result in a change of hydrological drought conditions, where response times are long enough or where a change in variability is caused by changes in the mean or the wet climate and recharge extreme. Hence, assessments of potential changes regarding average conditions or variability may have minor or no information for proactive drought planning. Our results suggest that drought assessments directly relevant for specific stakeholders' needs and analysed in the context of the local sensitivity determined by hydrogeological conditions will better allow for adaptation and planning.

The hydrogeological conditions are also linked to the locally specific precipitation accumulation time that has the maximum correlation with water table variation $T_{max}$. Hellwig et al. (2020) analysed the $T_{max}$ ranging from few months to several years across Germany. Their results suggested that $T_{max}$ can be a good proxy for heterogeneous reactions of the groundwater to droughts. The patterns of $T_{max}$ were similar to those found here for the groundwater's response to the more specific stress tests, hence the propagation time from meteorological to groundwater anomalies also has the potential to be a predictor of the general

groundwater drought sensitivity to recharge stress tests.

The drought-specific stress test modelling, however, provides a more nuanced insight into the hazard. The results for both $S_{SHIFT}$ and $S_{EVENT}$ revealed systematic differences for groundwater heads and baseflow. The main reason here is the non-linear relationship between the two variables: the baseflow dynamics are mainly driven by groundwater fluctuations in the wet range, when groundwater heads are closer to the surface and more groundwater discharge is possible through the dynamic drainage

network (Godsey and Kirchner, 2014). For low groundwater heads, the drainage system shrinks and less baseflow results in a lower sensitivity to changes in groundwater heads. In the model this is represented by the variable number of grid cells in a catchment that contribute to baseflow with less cells in case of low groundwater heads. Changes in groundwater heads due to the event stress tests are most pronounced in regions with long propagation times $T_{max}$ (taken from Hellwig et al., 2020) where the antecedent recharge has more influence. However, aquifers with long propagation times are usually characterized by large

dynamic storages leading to a smaller baseflow variability (i.e. more stable flow regimes). Correspondingly, large changes of baseflow occur predominantly in regions with short $T_{max}$ opposite to the regions of large groundwater head change.

The different responses of baseflow and groundwater are important to consider for an effective water management and drought planning in a changing climate. Different stakeholders will face different challenges in future and use the stress tests differently to design adaptation or to plan mitigation measures for emergency plans. For example, in a climate with higher annual recharge

sums but more frequent or severe summer droughts groundwater droughts might become less severe while the baseflow drought hazard becomes more severe. Where possible, one option might be to switch or add water use from surface water to groundwater to meet water demands for irrigation, industry, and public water supply. For other purposes relying on a minimal amount of surface water (e.g. navigation, water quality, or ecosystem health) adaptations such as regional water transfers or increased surface water storage capabilities might be more expedient.

## 5.2 Uncertainties of large-scale groundwater simulations under climate stress


The model used in this study is limited in that it simulates groundwater head and baseflow dynamics under natural conditions only. The usual anthropogenic response to drought is an increased groundwater pumping, which causes a positive feedback which accelerated drying (Famiglietti, 2014). Therefore, anthropogenic influences also need to be considered as significant contributors to real changes in groundwater heads (Kløve et al., 2014). Moreover, there is uncertainty arising from the aquifer

parametrization. Exact model derived $T_{max}$ as well as groundwater and baseflow drought severity must be taken with care and should not be interpreted exactly to the location. In particular, Hellwig et al. (2020) found a decreasing model performance for higher elevation regions with small scale variability of the hydrogeology. Gleeson et al. (2020) conclude in their commentary that profound (observation-based) model evaluations for large-scale groundwater models are currently beyond reach. Groundwater head dynamics measured at boreholes can deviate considerably from grid cell averages due to a large subgrid

heterogeneity (e.g. Kumar et al., 2016). Opposingly, baseflow dynamics can be seen as an integrated spatial signal but uncertainties arising from the separation of baseflow from streamflow are large (e.g. Stoelzle et al., 2020a). Also, for other observational data there are severe constraints (Gleeson et al., 2020). To allow for an effective local water management and reliable stress test results on this scale it will be most relevant to improve model parametrization with better hydrogeological data. However, even though the model uncertainties limit the use of model outputs on a local scale, they do not affect the

general conclusions on regional groundwater sensitivity found.

Climate change projections contain considerable uncertainties about future precipitation and predictions for recharge are even more uncertain as it might change even more strongly (Ng et al., 2010; Taylor et al., 2012; Jing et al., 2020). Studies on recharge changes in Central Europe consistently predicted increases during winter and decreases during summer (Eckhardt and Ulbrich, 2003; Stoll et al., 2011; Dams et al., 2012; Hunkeler et al., 2014; Chen et al., 2018), however, recharge is variable

with potentially large year-to-year variations (Kopp et al., 2018). As the magnitude of change is uncertain, the general sensitivity of a system as investigated in this study can help to assess, whether and where the expected contrasting seasonal change has a general potential to influence baseflow and groundwater drought. To further guide stakeholders according to their specific needs it can be beneficial to adopt stress test design and evaluation metrics or to complement stress tests with climate change projections.

There is evidence that different hydrometeorological characteristics that might change in future are relevant for groundwater and baseflow drought. Bloomfield et al. (2019) demonstrated an influence from changes in evapotranspiration due to increasing temperatures on changes in groundwater drought. Longobardi and Van Loon (2018) showed that changes in dry spell length can alter groundwater contributions to streamflow. Applying recharge frequency analysis to derive a 50-year or 100-year recharge drought event extrapolating beyond the range of the observational time period is a pragmatic hydrological design

concept. As always, it comes with uncertainty and may be questioned due to climate-change induced non-stationarity. But as a sensitivity testing framework, it is found useful and suitable for communication to practitioners used to dealing for example with flood frequency terminology. The $S_{EVENT}$ for the first-time provides country-scale composite estimates of groundwater

and baseflow sensitivity to such assumed more severe recharge droughts and should also be considered for future water management plans.

## 5.3 Benefits of complementary stress testing for sensitivity assessments

The different stress tests are complementary to modelling chains from climate change scenarios to hydrogeology as they target the groundwater's sensitivity against different characteristics that are important to consider for water management. $S_{SHIFT}$ focusses on systematic intra-annual changes in the recharge regime and its consequences for droughts. $S_{EVENT}$ assesses the specific response to prolonged dry spells whereas $S_{RECOV}$ investigates the groundwater's ability to recover after a severe drought. With the combination of these different stress tests different aspects of the groundwaters' sensitivity can be assessed and the following main points regarding the baseflow and groundwater drought sensitivity emerge:

1. Changes in the annual average recharge sum alter the groundwater heads in regions with slow groundwater response over the entire year, mitigating (or exacerbating if annual recharge is reduced) the groundwater drought hazard here for all seasons. In regions with fast groundwater responses, intra-annual recharge trends are more relevant than changes of the annual recharge sum.

2. An intra-annual shift of the recharge as it was assumed in $S_{SHIFT}$ has larger effects on baseflow and groundwater under average conditions than on water availability during drought. The general increase in baseflow and groundwater variability following stronger recharge seasonality does not necessarily result in a change of hydrological drought conditions.

3. Baseflow and groundwater respond to recharge on characteristic time scales. Hence, reduced antecedent recharge over a longer duration which could be a result of a changed climate with prolonged dry spells can lead to much more severe droughts in aquifers and surface waters reacting on the corresponding time scales.

4. Groundwater recovery times for a severe drought are mainly related to the hydrogeology. This finding supports recent approaches for predictions on groundwater drought development several months ahead based on the site-specific characteristics of groundwater dynamics (e.g. Prudhomme et al., 2017; Parry et al., 2018).

## 6 Conclusions

Future changes of recharge are relevant for the groundwater drought hazard and groundwater's potential to mitigate drought impacts. In this study a stress test approach was employed to test the groundwater's system sensitivity to changes in recharge: three generic recharge stress tests were used in a country-scale German groundwater model simulating groundwater heads and baseflow. Different from climate change scenarios, the stress tests systematically apply different types of recharge change (e.g. proportional shifts or extreme events of a given return period) allowing for general conclusions on the diversity of groundwaters' sensitivity. While the assumed intra-annual recharge shifts can be expected to weaken the groundwater drought

hazard, prolonged dry spells may aggravate droughts, particularly in regions with slow responding aquifers. Baseflow is not linearly related to changes of groundwater heads and is more prone to intensified drought event conditions on a shorter time scale, especially in regions with fast responding aquifers. The groundwaters' drought recovery time is strongly related to the aquifers' characteristic response time scale. Hence, spatial patterns of recovery times are only secondarily depending on the meteorological drought characteristics but rather an inherent property of the aquifer with large regional differences.

The stress test approach applied in this study allows for a detailed composite assessment of a controlled environmental change. Regional sensitivities to changes in recharge differ considerably. Hence, key regions most vulnerable to recharge changes can be identified and may enable proactive adaptations for different stakeholders independent of specific climate projections. Different regional sensitivities could also be used for probabilistic real-time groundwater drought forecasting as an informative tool for water supply and other stakeholders. While recently developed country-to-global scale transient and gradient-based groundwater models can guide decision-making on these scales, for local management decisions it will be important to consider local hydrogeological conditions and include also anthropogenic feedbacks such as increased pumping during drought (e.g. due to higher irrigation demand). Such feedback could be also implemented as generic stress tests. Therefore, future work evaluating the groundwater response to scenarios of human water use during drought will be needed to complement the findings of this study.

**Data availability**

The model outputs from the reference run and stress test runs can be downloaded from FreiDok (https://doi.org/10.6094/UNIFR/167379, will be activated upon acceptance).

**Author contribution**

JH developed the main ideas, the design of the stress tests was jointly developed by all authors. JH performed the analyses and prepared the manuscript which was reviewed by the co-authors.

**Acknowledgements**

We acknowledge the comments by Editor Jim Freer and two anonymous reviewers who provided valuable reviews which significantly helped to improve the quality of the paper. JH was funded by the DFG project TrenDHy STA632/4-1 and KS by the DFG Heisenberg programme STA632/3-1, MS by the LUBW grant "Low Flow Stress Tests".

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

**Table 1: Overview of the three generic stress tests used in this study and the related question to be answered by the stress test.**

|  | Question to be answered | Time frame | Boundary conditions | Recharge modifications |
|---|---|---|---|---|
| $S_{SHIFT}$ | How will a changed recharge regime with wetter winters and drier summers change the inter-annual variability and water availability during droughts? | Corresponding to reference simulation (57 years) | Apart from recharge same as for the reference simulation | Winter decrease, summer increase of different strength ($\pm$ 5, 10, 15, 20, 30 % relative to reference simulation) |
| $S_{EVENT}$ | Could the effect have been worse? How sensitive are hydrological droughts to antecedent recharge conditions on different durations? | Historical events | Taken from the historical event in the reference simulation | Recharge from reference simulation rescaled to match a drought event with a return period of 50 (100) years for three different durations |
| $S_{RECOV}$ | What is the recovery time needed to terminate a severe drought event? | Hypothetical event | Most severe drought modelled in the reference simulation for every grid cell taken as initial conditions | Long-term monthly average/ 25-percentile/ 75-percentile recharge from the reference simulation |

**Table 2: Required model ability and discussion of model performance for the different stress tests.**

| | Required model ability | Evaluation metric | Model performance assessment |
|---|---|---|---|
| $S_{SHIFT}$ | Reliable propagation of inter- and intra-annual recharge dynamics into groundwater heads and baseflow | $T_{max}$ | Overall, the model depicts both, differences of $T_{max}$ across the study area and the systematically shorter $T_{max}$ of baseflow compared to groundwater. However, for baseflow $T_{max}$ was notably overestimated in the North and underestimated in the South while for groundwater it was overestimated in the porous aquifers of the lowlands and underestimated in higher elevations (see Hellwig et al., 2020 for more detailed analyses). Hence, absolute $S_{SHIFT}$ responses may be biased in that same way. The model estimates allow for highest confidence in the representation of general shift-patterns across the study area. |
| $S_{EVENT}$ | Reliable model representation of benchmark drought events | Differences between observed and modelled groundwater /baseflow drought severities | Simulations and observations show a considerable variability of groundwater drought severity for different drought years across the study area. Consistent with observations, modelled drought severities were weaker in 2003 compared to 1973 with several regions in the study area not in groundwater drought. These patterns are also consistent with state agency reports (see Hellwig et al., 2020). However, especially in the Northeast the model responds too slowly (corresponding with too long $T_{max}$, see above) leading to deviating groundwater drought severities: the drought severity of 1973 is overestimated in the model while it is underestimated for 2003. For baseflow model performance is similar: while general patterns of drought severity can be depicted, drought severities deviate most in the North (-East) (see also Figure S1). Overall, there are systematic uncertainties arising from the comparison of observational data with model outputs which might relate to some of the differences found (for a more advanced discussion on that see Hellwig et al., 2020, Section 2.3). |
| $S_{RECOV}$ | Reliable representation of severe drought + propagation of recharge forcing into groundwater | Combination of evaluation metrics of $S_{SHIFT}$ and $S_{EVENT}$ | As both general patterns of drought severities and the propagation of the forcing into groundwater are captured by the model, prerequisites for an appropriate drought termination simulation are given. Uncertainties for this test are – similar to the other stress tests – largest in regions of weaker model performance regarding $T_{max}$. |


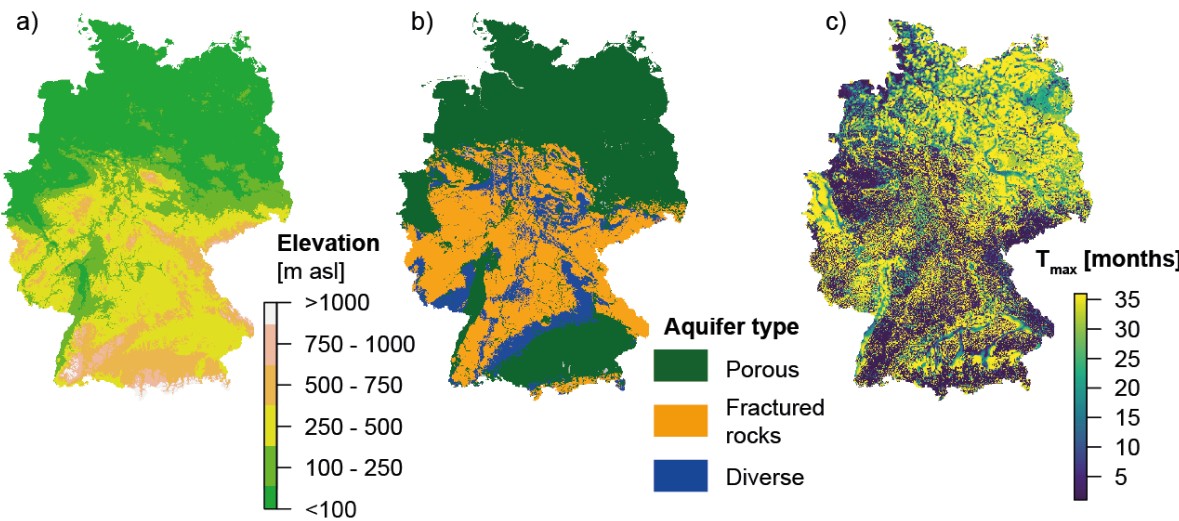

**Fig. 1:**   **Study area. a) Topographic map, b) main aquifer types (taken from BGR and SGD, 2016) and c) precipitation**
**accumulation times that have the maximum correlation with groundwater $T_{max}$ (taken from Hellwig et al., 2020).**

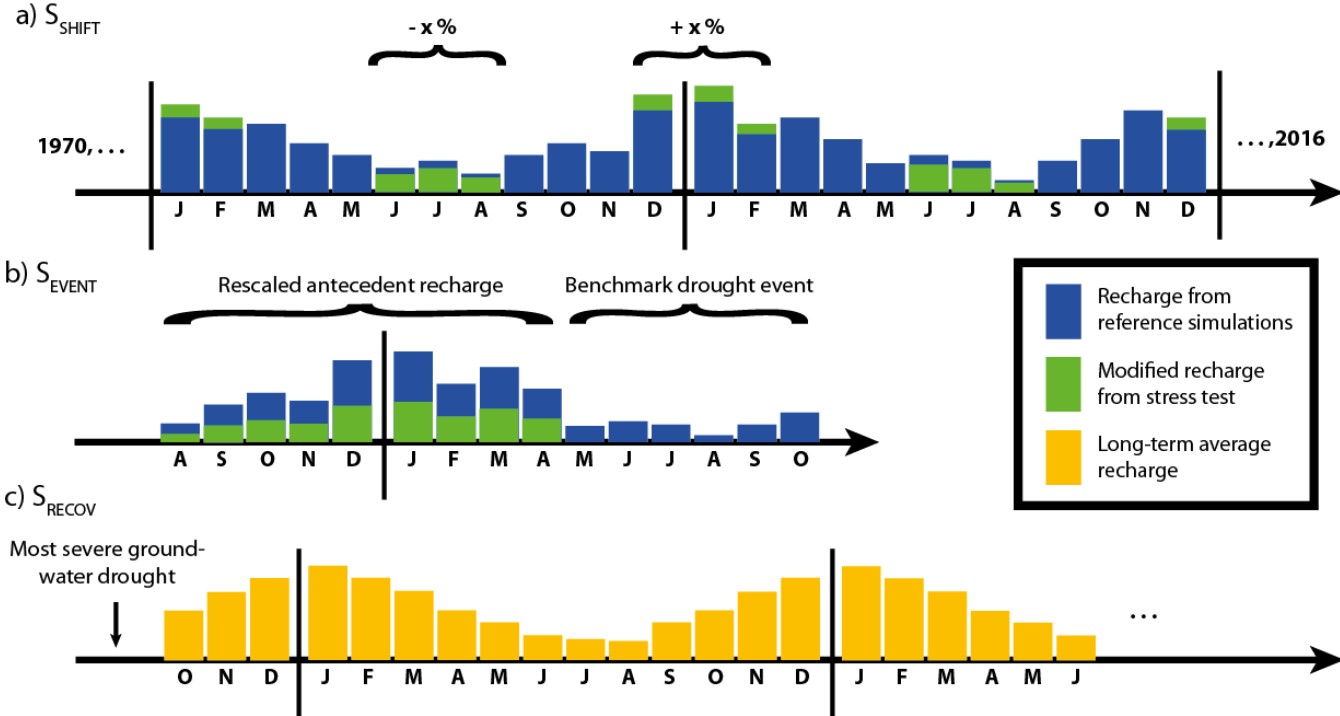

Fig. 2: Recharge modifications used for the three different stress test types in this study.


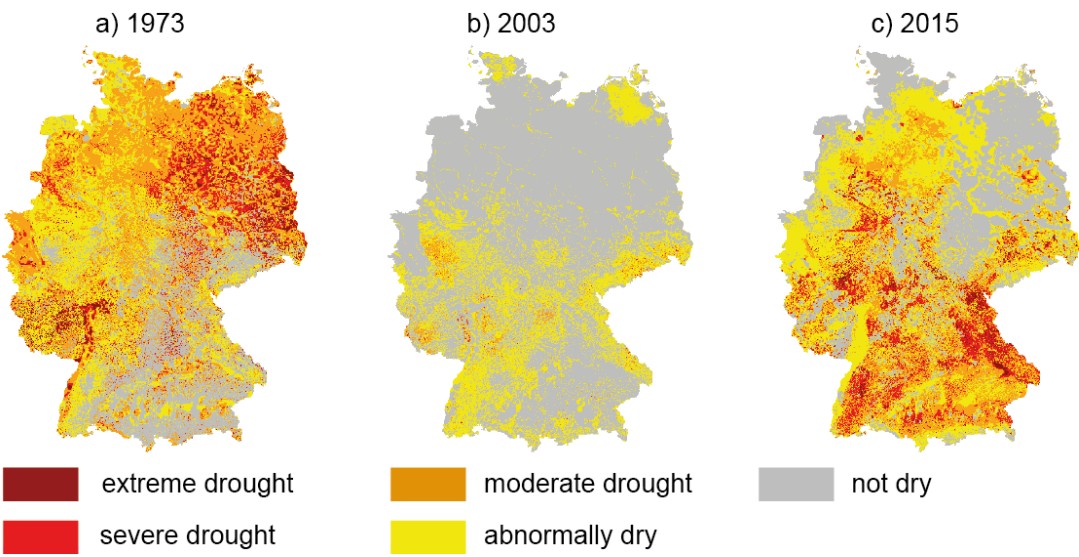

**Fig. 3:** **Modelled groundwater drought situation during summer months (JJA) for benchmark drought events. Drought classes are derived from average standardized water table referring to the thresholds -2 (2.3 % of time: extreme drought), -1.5 (6.7 % of time: severe drought), -1 (15.9 % of time: moderate drought) and 0 (50 % of time: abnormally dry).**


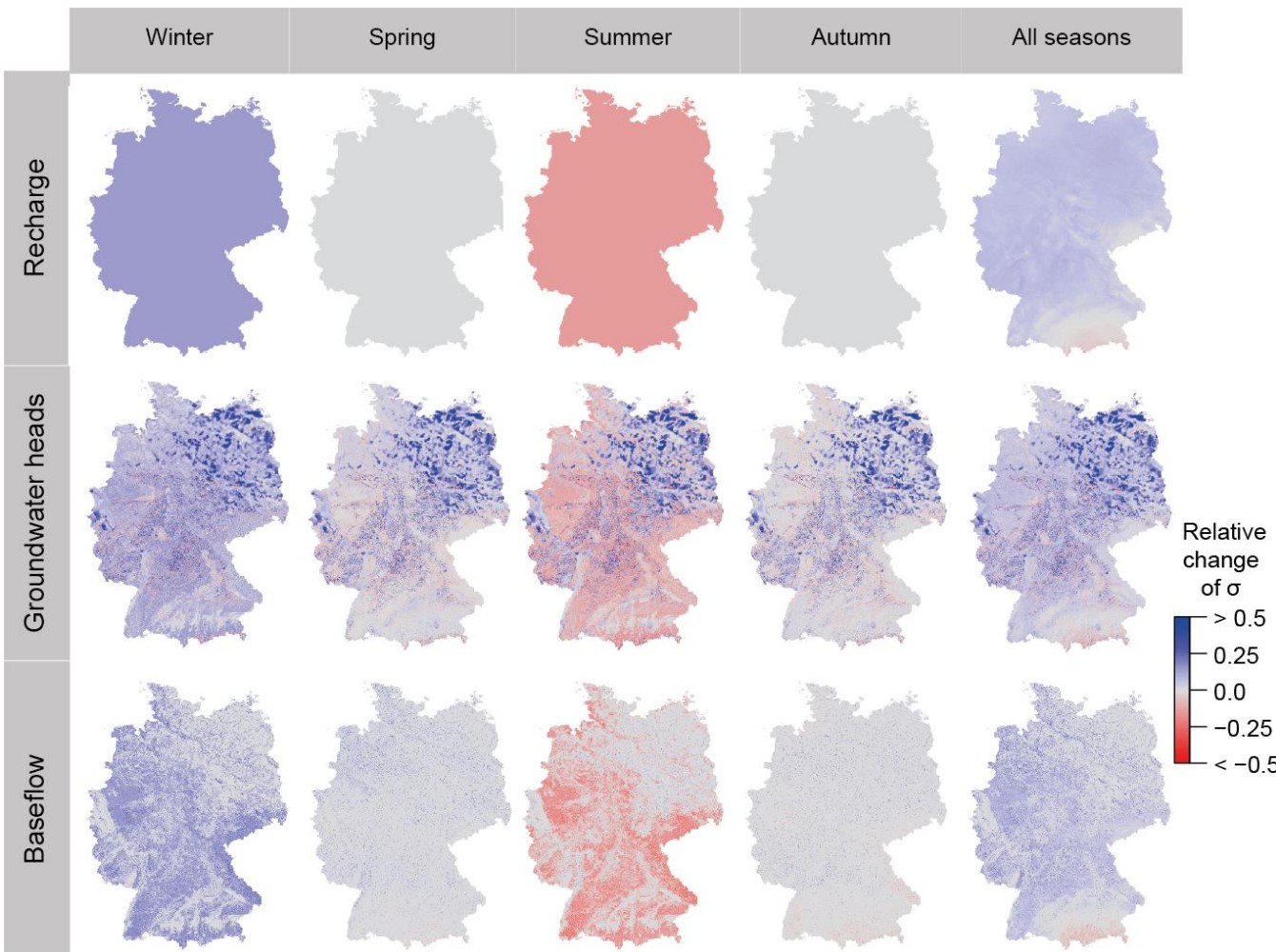

**Fig 4: Relative changes in the inter-annual variability of recharge, groundwater head and baseflow for different seasons (with winter: DJF, spring: MAM, summer: JJA, autumn: SON) and a seasonal shift of 15%.**


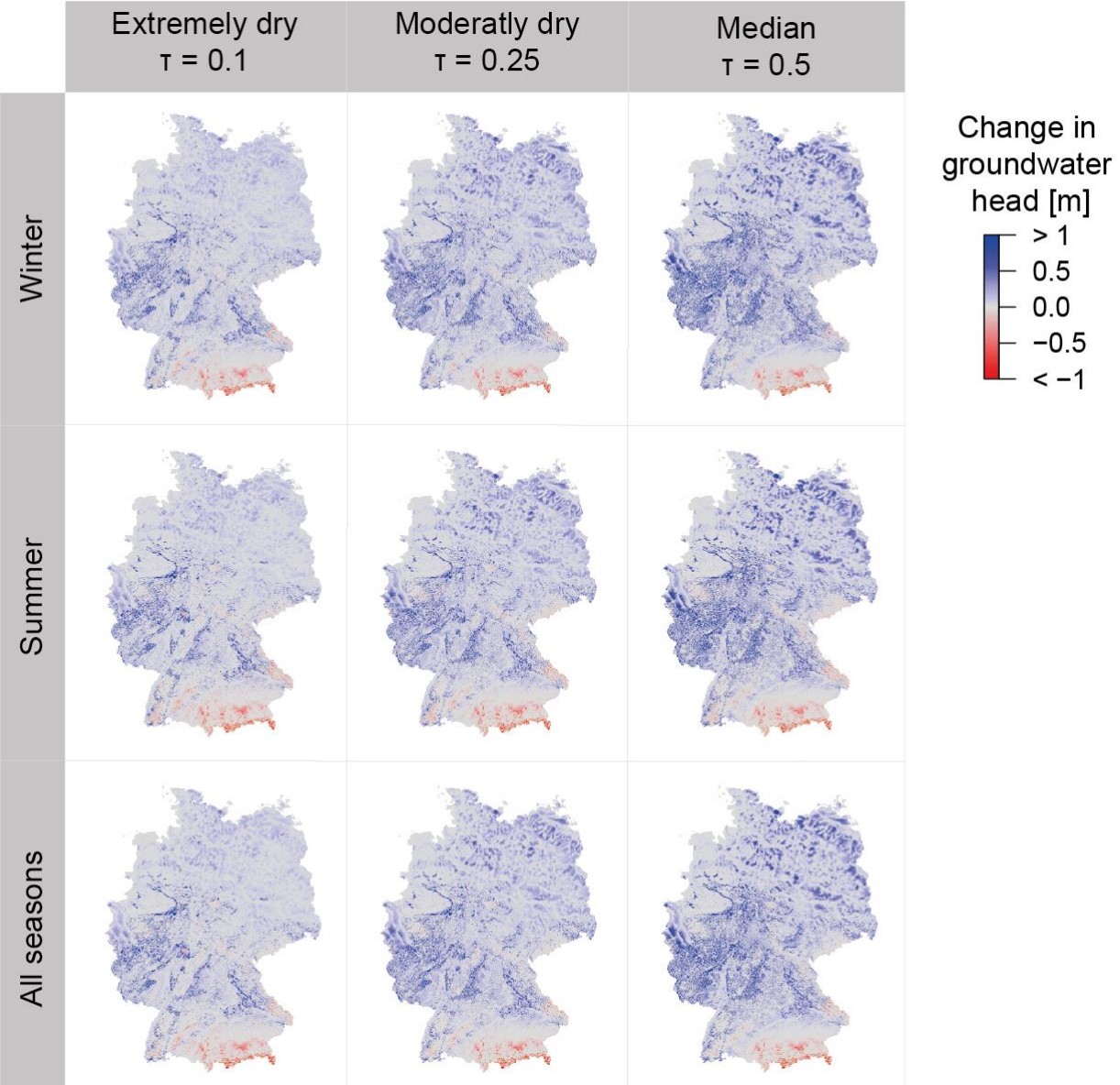

**Fig. 5: Groundwater head changes for S**<sub>SHIFT</sub> **in Germany for selected drought thresholds (columns) for different seasons (rows) for a shift of 15%.**

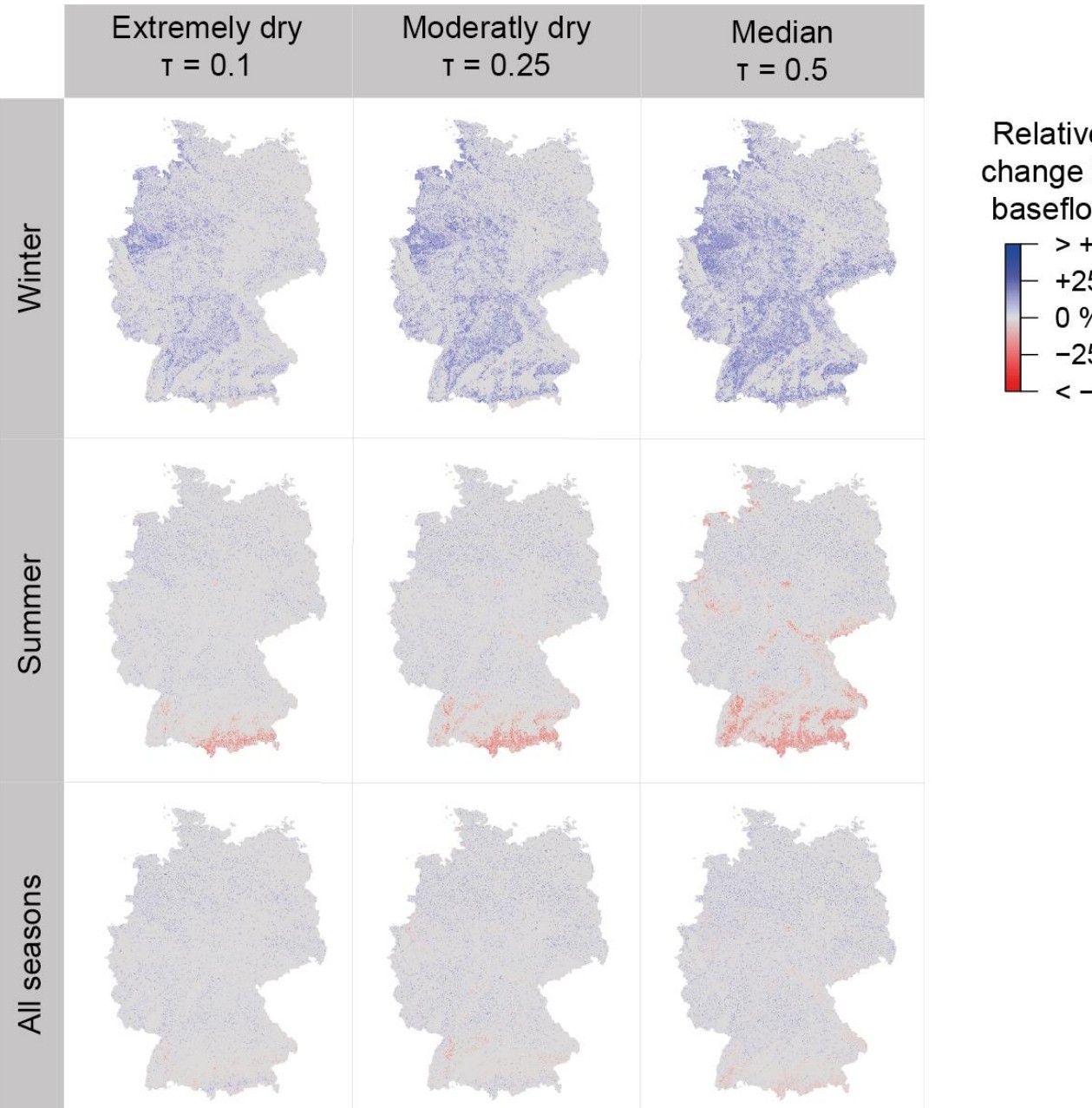


**Fig. 6: Same as Figure 5 for relative changes of baseflow.**

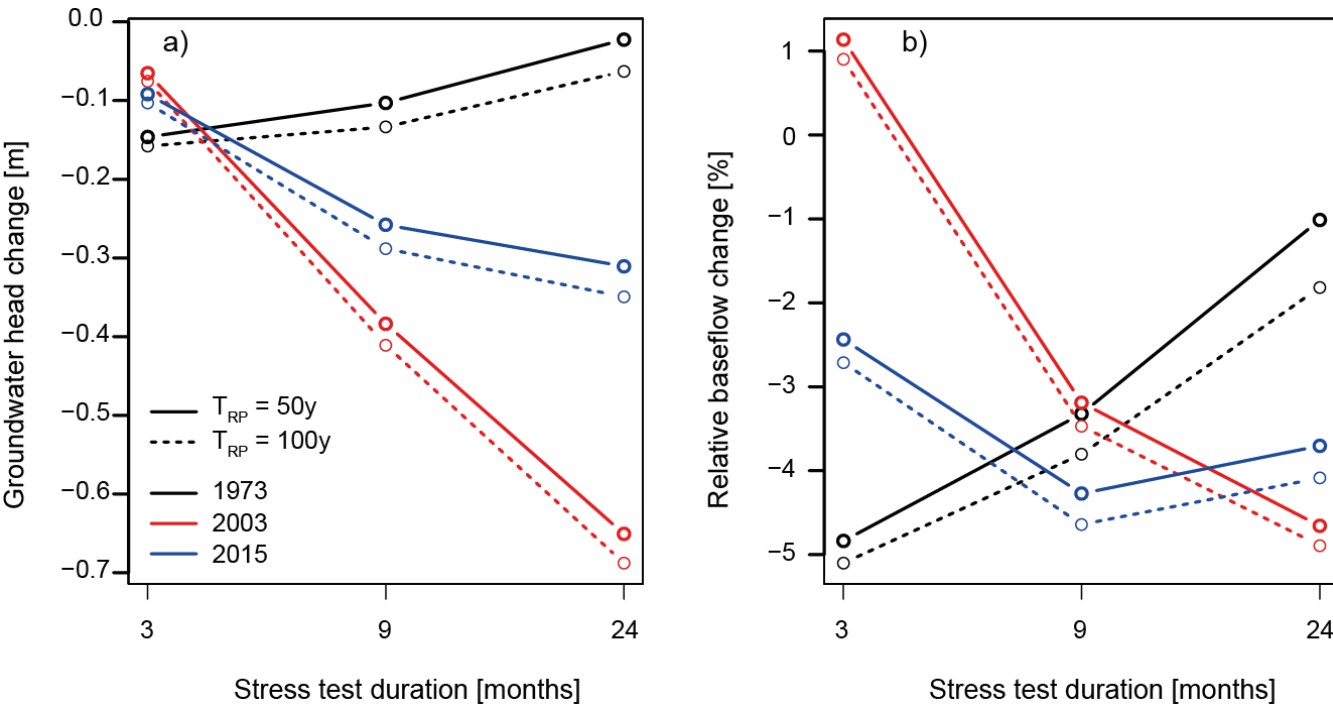

**Fig. 7: Changes during drought averaged over Germany for all different S$_{EVENT}$ stress tests: response of different events (1973, 2003,**
**2015), different antecedent recharge reduction time scales (3, 9, 24 months) and two return periods (T$_{RP}$ = 50 and T$_{RP}$ = 100 years).**
**a) Groundwater head changes, b) relative baseflow changes.**

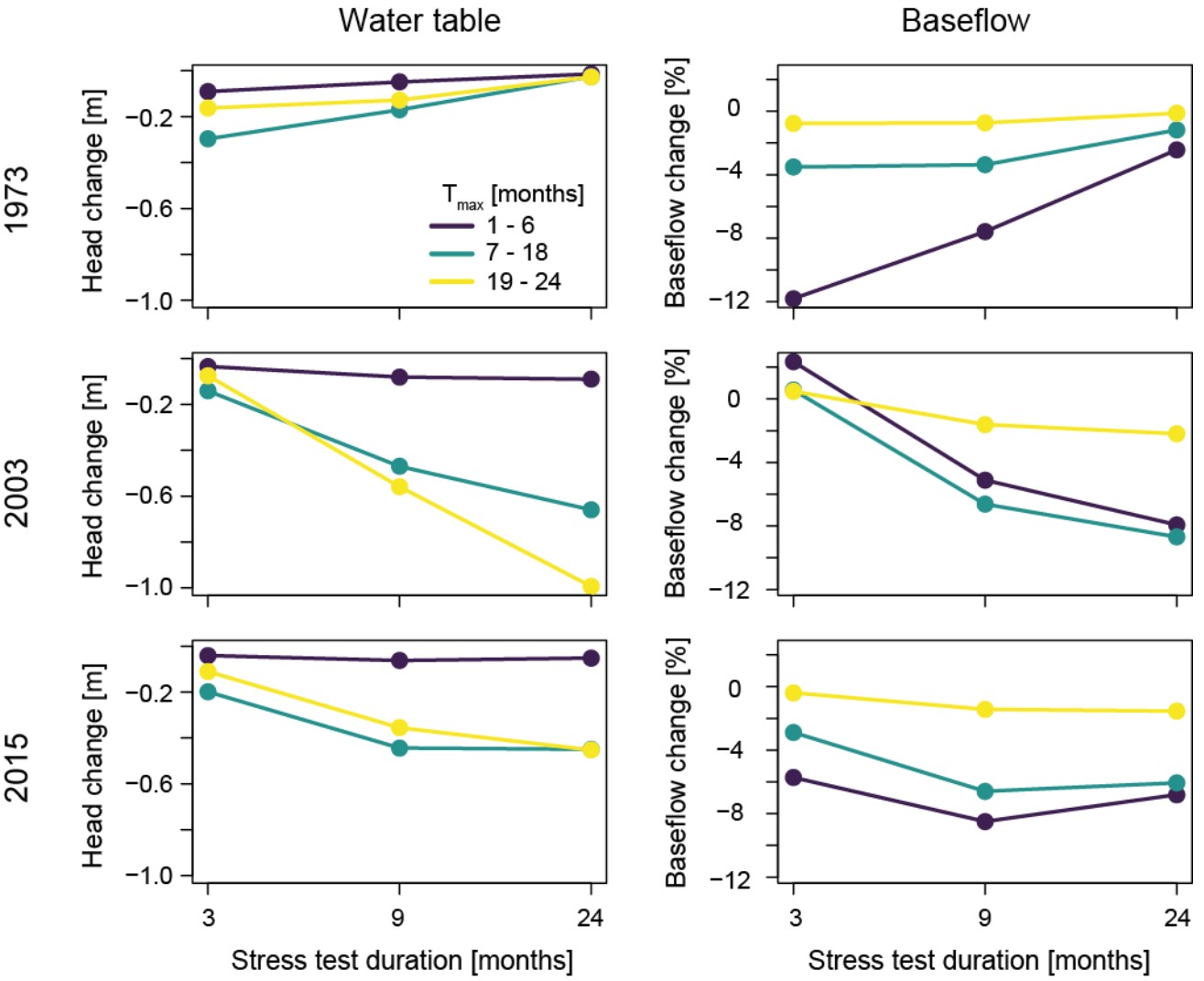

**Fig. 8: Effects of S𝐄𝐕𝐄𝐍𝐓 with T𝐑𝐏 = 50y for three different classes of $T_{max}$ averaged over Germany. Note the different scales for the y-axes.**

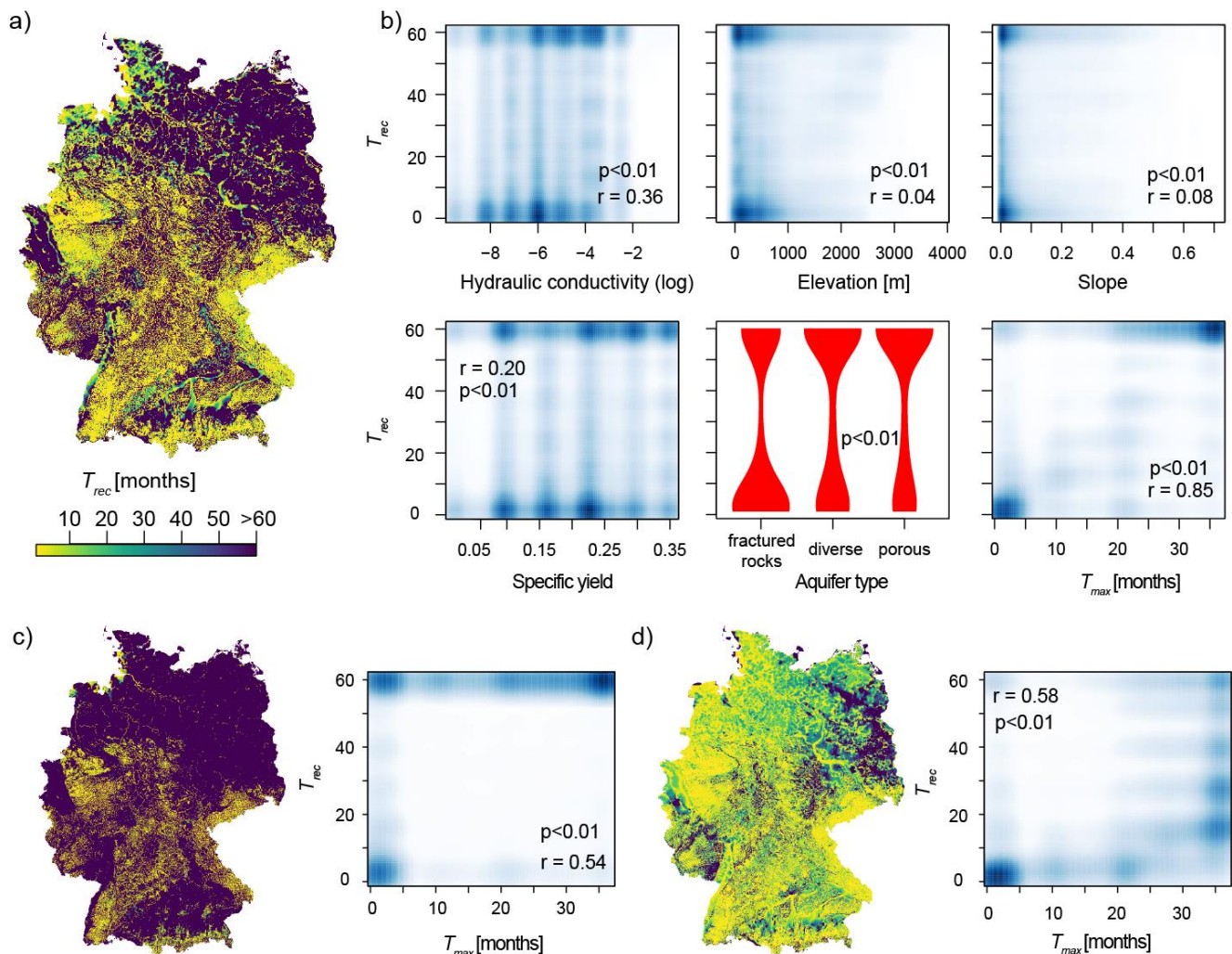

**Fig. 9: Recovery time $T_{rec}$ for $S_{RECOV}$. a) spatial distribution of $T_{rec}$ across Germany, b) relationship between $T_{rec}$ and model parameters hydraulic conductivity, elevation, slope and specific yield, aquifer type (taken from HÜK200) and precipitation accumulation time that has the maximum correlation with groundwater $T_{max}$ (taken from Hellwig et al., 2020). c) + d) spatial distribution of $T_{rec}$ and $T_{rec}$ over $T_{max}$ for dry resp. wet conditions during drought recovery. Blue colours indicate the smoothed density derived from all model grid cells. Red violins illustrate the distribution of $T_{rec}$ in three different categories of aquifer type. r is the Pearson correlation coefficient for the variables compared, p is the corresponding p-value.**
