# Peer review of "Groundwater and baseflow drought responses to synthetic recharge stress tests"

_Hydrology and Earth System Sciences, 2020_

## Referee Comment (RC1) · Anonymous Referee #1 · 12 Aug 2020

The paper describes a study where a large-scale, high-resolution MODFLOW-groundwater model of Germany has been used to assess a range of potential changes to groundwater and baseflow drought hazard based on three change scenarios. The scenarios are: i.) a changed recharge regime with wetter winters and drier summers (SSHIFT), ii.) changes to antecendent conditions associated with three major historic episodes of drought in Germany (SEVENT), and iii.) recovery from drought (SRE-COV). These scenarios were co-designed in part with the Climate and Water Initiative of southern Germany's federal states (KLIWA) (L67-86) with the aim of stress testing the sensitivity to drought of groundwater and baseflow. Although, the geographical focus of the study is Germany, the paper addresses questions relevant to a wide read-

ership and is clearly in the scope of HESS.

The description of the model setup (Section 2) is adequate given that more details can be found in the paper by Hellwig et al. (2020) who developed the model. However, a critical assessment by the authors of the models suitability, including method of calibration and appropriateness of it's underlying assumptions, for the current application would be helpful. The description of the scenario design and modelling approach (Section 3) is generally clear and well-reasoned. However, the scenarios appear somewhat arbitrary. In particular, the formulation of the SRECOV scenario is less convincing than the other two scenarios. To assess the maximum duration for groundwater recovery from severe drought, the lowest simulated groundwater heads are taken as an initial condition and groundwater heads are simulated using long-term average monthly recharge as input until an arbitrary recovery has been achieved. Although adequately described, the motivation and justification for the details of this scenario are not given.

The results are presented well, both graphically and in their description in Section 4. The Discussion provides a number of interesting insights into the results. For example, the authors make the observation at L287-290 that: "the different responses of baseflow and groundwater are important to consider for an effective water management in a changing climate. For example, in a climate with higher annual recharge sums but more frequent summer droughts groundwater droughts might become less severe while the baseflow drought hazard becomes more severe with potential impacts on economy and ecology". Given that the scenarios that led to this observation were shaped by stakeholders, it would be interesting to know if and how stakeholders might use such information. More generally, given the nature of the set-up of the paper (e.g. L67-74) it would be interesting to hear the author's views on any specific implications of the results of their study for drought planning and management. These could be described, however briefly, in the Discussion.

Specific comments: L229 I think that the authors meant "relative" not "relevant"?

Section 4.1. The authors make a number of observations relating to the groundwater and baseflow changes being more pronounced under average conditions than for drought, and this is also highlighted in the Summary at L326. A brief interpretation and discussion of the implications of these observations would be helpful.

Section 4.3 and Figure 9. The main feature of the analysis of recovery time appears to be the essentially bi-modal nature of Trec, this being most evident in the Trec v Tmax plot in Fig. 9. It would be interesting to hear what the authors think might be contributing to this result. Does it reflect intrinsic characteristics of the modelled system, is it an artefact of the model structure or calibration, or is it some combination of both? Perhaps such a discussion could be added to Section 4.3?

L348-350. The first and only mention of the application of this approach to is in the Conclusions. This seems strange. IT may be appropriate to include these observations in the Discussion, but not in the conclusions?

---

## Author Comment (AC1) · 18 Aug 2020

**We would like to thank Referee #1 for the positive feedback and helpful suggestions on this manuscript. Below we give point-by-point responses to the comments (bold and italic).**

1) The paper describes a study where a large-scale, high-resolution MODFLOW-groundwater model of Germany has been used to assess a range of potential changes to groundwater and baseflow drought hazard based on three change scenarios. The scenarios are: i.) a changed recharge regime with wetter winters and drier summers (SSHIFT), ii.) changes to antecedents conditions associated with three major historic episodes of drought in Germany (SEVENT), and iii.) recovery from drought (SRECOV). These scenarios were co-designed in part with the Climate and Water Initiative of southern Germany's federal states (KLIWA) (L67-86) with the aim of stress testing the sensitivity to drought of groundwater and baseflow. Although, the geographical focus of the study is Germany, the paper addresses questions relevant to a wide readership and is clearly in the scope of HESS.

The description of the model setup (Section 2) is adequate given that more details can be found in the paper by Hellwig et al. (2020) who developed the model. However, a critical assessment by the authors of the model's suitability, including method of calibration and appropriateness of it's underlying assumptions, for the current application would be helpful. The description of the scenario design and modelling approach (Section 3) is generally clear and well-reasoned. However, the scenarios appear somewhat arbitrary. In particular, the formulation of the SRECOV scenario is less convincing than the other two scenarios. To assess the maximum duration for groundwater recovery from severe drought, the lowest simulated groundwater heads are taken as an initial condition and groundwater heads are simulated using long-term average monthly recharge as input until an arbitrary recovery has been achieved. Although adequately described, the motivation and justification for the details of this scenario are not given. The results are presented well, both graphically and in their description in Section 4.

*The national-scale groundwater model is certainly limited in its local validity. We reflect on the limitations regarding several aspects in the Discussion Section. However, we agree that another reflection on the suitability of the model for the scenarios applied will be useful and will add this to the model and/or scenario description together with further details on model calibration etc.*

*We agree that the formulation of SRECOV might appear arbitrary. We selected the scenario to address the stakeholders request for a better understanding of the drought termination period. To account for the local differing conditions, we adopted the idea of a 'composite map' and did not select one specific drought year but rather the lowest heads of the simulated period. We agree that the recovery threshold is arbitrary, however, additional analyses with other thresholds (40 and 50-percentile groundwater head) resulted in the same patterns. As we find that the resulting $T_{rec}$ are strongly related to $T_{max}$, which is an aquifer characteristic independent of meteorology/recharge or initial conditions, it is reasonable that our choices regarding initial conditions and recharge are not substantially relevant for the results. We will discuss these points in the revised manuscript and rephrase the motivation to make clear that we use this scenario to learn about the general time scale of recovery from a severe drought and exact values of $T_{rec}$ depend on initial conditions, assumed recharge and recovery threshold.*

2) The Discussion provides a number of interesting insights into the results. For example, the authors make the observation at L287-290 that: "the different responses of baseflow and groundwater are important to consider for an effective water management in a changing climate. For example, in a climate with higher annual recharge sums but more frequent summer droughts groundwater

droughts might become less severe while the baseflow drought hazard becomes more severe with potential impacts on economy and ecology". Given that the scenarios that led to this observation were shaped by stakeholders, it would be interesting to know if and how stakeholders might use such information. More generally, given the nature of the set-up of the paper (e.g. L67-74) it would be interesting to hear the author's views on any specific implications of the results of their study for drought planning and management. These could be described, however briefly, in the Discussion.

*Both baseflow and groundwater deficits can have negative impacts, but the affected sectors and planning tools can be quite different. In Germany, public water supply relies to a large proportion on groundwater resources (economic/social impact). SSHIFT will help to make water supply resilient, e.g. by diversifying sources with different seasonal sensitivity; SEVENT may be used to test water tower storages at extreme event scale together with SRECOV to plan durations of measures that may be necessary. Additionally, groundwater dependent ecosystems can be vulnerable to groundwater droughts (ecological impact). Baseflow droughts also affect surface water quality and water ecology (ecological impact) as well as navigation, energy production and tourism/recreation (economic/social impacts). Therefore, different stakeholders will face different challenges and use the scenarios differently to design adaptation or to plan mitigation measures for the emergency. We will name some examples in the revised version. Also, the combination of the two variables studied may be useful. In case of increased baseflow drought hazard but less change in groundwater drought hazard one option might be to switch or add – as far as practicable - water use from surface water to groundwater. For agricultural irrigation this process has already taken place in some parts of the study region, e.g the Upper Rhine valley.*
*We will discuss these points in the revised manuscript.*

3) Specific comments: L229 I think that the authors meant "relative" not "relevant"?

*Agree*

4) Section 4.1. The authors make a number of observations relating to the groundwater and baseflow changes being more pronounced under average conditions than for drought, and this is also highlighted in the Summary at L326. A brief interpretation and discussion of the implications of these observations would be helpful.

*Agree. In particular it is interesting to see, that an increased variability does not necessarily lead to more pronounced extremes. We will add a brief interpretation to the Discussion Section.*

5) Section 4.3 and Figure 9. The main feature of the analysis of recovery time appears to be the essentially bi-modal nature of Trec, this being most evident in the Trec vs. Tmax plot in Fig. 9. It would be interesting to hear what the authors think might be contributing to this result. Does it reflect intrinsic characteristics of the modelled system, is it an artefact of the model structure or calibration, or is it some combination of both? Perhaps such a discussion could be added to Section 4.3?

*Yes, the bi-modal nature of $T_{rec}$ is strongly related to $T_{max}$. The large differences in $T_{max}$ with some regions responding within few months and others over several years are a characteristic we can also find in observations (e.g. see Figure 7 in Hellwig et al., 2020), so we don't think it is just an artefact of the modelling. Anyway, the model does not capture Tmax equally well everywhere. For example, in Hellwig et al. (2020) it was noted that $T_{max}$ was "overestimated by the model in the*

*porous aquifers in the lowlands and underestimated in higher elevations". Certainly, this will also influence the pattern of Trec. We will reflect on this in the Discussion Section.*

6) L348-350. The first and only mention of the application of this approach to is in the Conclusions. This seems strange. It may be appropriate to include these observations in the Discussion, but not in the conclusions?

*Agree*

---

## Referee Comment (RC2) · Anonymous Referee #2 · 28 Aug 2020

**Review of Hellwig et al (2020), Stress-testing groundwater and baseflow drought responses to synthetic climate change-informed recharge scenarios**

This paper tackles an important topic of how groundwater and baseflow will respond to changes in recharge. To test this, the study uses MODFLOW to explore how groundwater and baseflow change in response to three different recharge scenarios across Germany. The recharge scenarios are informed from stakeholder interactions and the combination of the scenarios targets different characteristics of groundwater and baseflow drought responses. The study concludes that a shift in rainfall to wetter winters and drier summers will not cause decreases in groundwater resources in general, but water managers need to consider the potential for more severe groundwater droughts following prolonged dry spells. The figures are well presented and the paper is generally well written.

The results could be of significant interest to the scientific community. However, my overall assessment is that major changes to the paper with additional simulations are required before the paper is suitable for publication. Currently the paper explores a very limited set of scenarios and thus does not robustly "stress test" or truly assess the sensitivity of groundwater and baseflow drought responses to different scenarios. It is difficult to have confidence in the conclusions that are presented in the paper when they are based on a single change for each scenario. This becomes particularly important given the significant non-linearities between changes in groundwater head and baseflow, as highlighted by the authors. A critical assessment of the model's suitability to simulate groundwater and baseflow drought responses is also needed.

These comments are discussed in more detail below, which I hope the authors find useful.

**Major Comments**

**Scenarios** – The scenarios are very limited. If the aim of the paper is to test and attribute specific sensitivities as noted in the introduction then a larger number of simulations should have been undertaken. Conclusions such as "a seasonal shift of recharge (i.e. less summer recharge and more winter recharge) will therefore have low effects on groundwater and baseflow drought severity" need to be based on more than a single scenario of +/-15% to be robust. Specific comments are:

(1) *SShift* – This scenario applies a 15% increase in recharge for winter months and a 15% decrease in recharge for summer months to the whole time series. Running a single set of percentage changes applied to the whole timeseries provides a very limited view of the question posed of "How will a changed recharge regime with wetter winters and drier summers change the inter-annual variability and water availability during droughts?". The authors should explore this in more depth by running additional scenarios that vary the percentage increases.

(2) *Srecov* – The justification for this scenario is quite weak compared to the other two scenarios and again is very limited in that it only explores the response under the assumption of long term average recharge.

(3) *Comparison between scenarios* – In the discussion and conclusions, comparisons between the scenarios are made. However, it is difficult to be confident in these comparisons as only a single scenario is assessed. For these comparisons to be robust additional simulations need to be performed to assess the sensitivity of the drought response to each scenario.

**Model Evaluation** – I agree with reviewer 1 that a critical assessment of the model's suitability for this application is required in Section 2. The authors need to demonstrate that the model can effectively reproduce the metrics that are used in this paper to assess groundwater and baseflow drought responses (e.g. the recovery time $T_{rec}$, inter-annual variability, percentile thresholds, performance during "benchmark droughts") and how this varies spatially and temporally for Germany. Currently, the discussion in Section 2 centres on model performance for $T_{max}$ which is

based on correlations and not focused on the (likely) non-linear drought responses that are being assessed here.

**Minor Comments and Technical Corrections**

**Abstract L7.** Please change to "depend on **the** systems' sensitivity"

**Introduction L25-28.** I would move (or remove) the two sentences starting with "Contrary to surface water, groundwater is hard to…" to L44 where you discuss the absence of observational data and use of groundwater models in more detail.

**Introduction L49.** Replace "more and more" with "increasingly"

**Introduction L55.** "Climate models (often) lack alterations in the sequencing of future wet and dry spells". This sentence needs to be supported by some references.

**Equation 1 L114.** What does the '$f$' denote?

**Section 3 L164.** It is not entirely clear to me how you calculate inter-annual variability – can you clarify and provide the equation?

**Discussion.** It might be worth adding some sub-section headings to the discussion to break up the text a little for the reader.

**Discussion L267-268 "**Also, the recovery time Trec from a severe drought varied accordingly (SRECOV)." I am not sure what you mean here – can you clarify?

**Supplementary Information.** Figures S1-S4 are very difficult to interpret and the figure quality is poor (i.e. they are quite blurry). Can you make the maps bigger and ensure the figures are incorporated at high resolution so that they are clear to the reader.

---

## Author Comment (AC2) · 2 Sep 2020

**We would like to thank Referee #2 for the feedback and helpful suggestions on this manuscript. Below we give point-by-point responses to the comments (bold and italic).**

1) This paper tackles an important topic of how groundwater and baseflow will respond to changes in recharge. To test this, the study uses MODFLOW to explore how groundwater and baseflow change in response to three different recharge scenarios across Germany. The recharge scenarios are informed from stakeholder interactions and the combination of the scenarios targets different characteristics of groundwater and baseflow drought responses. The study concludes that a shift in rainfall to wetter winters and drier summers will not cause decreases in groundwater resources in general, but water managers need to consider the potential for more severe groundwater droughts following prolonged dry spells. The figures are well presented and the paper is generally well written.

The results could be of significant interest to the scientific community. However, my overall assessment is that major changes to the paper with additional simulations are required before the paper is suitable for publication. Currently the paper explores a very limited set of scenarios and thus does not robustly "stress test" or truly assess the sensitivity of groundwater and baseflow drought responses to different scenarios. It is difficult to have confidence in the conclusions that are presented in the paper when they are based on a single change for each scenario. This becomes particularly important given the significant non-linearities between changes in groundwater head and baseflow, as highlighted by the authors. A critical assessment of the model's suitability to simulate groundwater and baseflow drought responses is also needed.

These comments are discussed in more detail below, which I hope the authors find useful.

***We acknowledge that the term 'stress-test scenario' might be misleading as 'scenarios' are often linked to ensembles of slightly different pathways. As our scenarios are considerably different from that, we will adopt the terminology in the revised manuscript to make this more transparent. With our stress-tests we specifically aim to meet stakeholders' requests for simple and easily interpretable scenarios that rather give information on possible general directions of change instead of uncertainty ranges depending on specific scenario assumptions. We will put this aim more clearly in the revised manuscript. Additionally, we will expand our model runs and the evaluation (details below).***

2) **Scenarios** – The scenarios are very limited. If the aim of the paper is to test and attribute specific sensitivities as noted in the introduction then a larger number of simulations should have been undertaken. Conclusions such as "a seasonal shift of recharge (i.e. less summer recharge and more winter recharge) will therefore have low effects on groundwater and baseflow drought severity" need to be based on more than a single scenario of +/-15% to be robust. Specific comments are:

(1) *SShift* – This scenario applies a 15% increase in recharge for winter months and a 15% decrease in recharge for summer months to the whole time series. Running a single set of percentage changes applied to the whole timeseries provides a very limited view of the question posed of "How will a changed recharge regime with wetter winters and drier summers change the inter-annual variability and water availability during droughts?". The authors should explore this in more depth by running additional scenarios that vary the percentage increases.

(2) *Srecov* – The justification for this scenario is quite weak compared to the other two scenarios and again is very limited in that it only explores the response under the assumption of long term average recharge.

(3) *Comparison between scenarios* – In the discussion and conclusions, comparisons between the scenarios are made. However, it is difficult to be confident in these comparisons as only a single scenario is assessed. For these comparisons to be robust additional simulations need to be performed to assess the sensitivity of the drought response to each scenario.

*The intention of our SShift and Srecov generic scenarios is to identify site-specific sensitivities to certain general hydroclimatic conditions which are of special interest to different stakeholders (in the revised manuscript we will phrase this more clearly). Typically, these sensitivities are much more driven by the physiographic and hydrogeological conditions compared to the exact climatic forcings tested. However, we agree that the results can become more reliable with a broader range of tested forcings (in our case directly recharge) and different responses of baseflow and groundwater can be analysed in more detail with more model runs. Therefore, for the revised manuscript we plan to run additional simulations of $S_{Shift}$ (assuming other percentages of change such as 5%, 10%, 20% and 30%) and $S_{Recov}$ (assuming rather wet and dry conditions during recovery). This will also help to better compare the outcomes of the different scenarios.*

3) **Model Evaluation** – I agree with reviewer 1 that a critical assessment of the model's suitability for this application is required in Section 2. The authors need to demonstrate that the model can effectively reproduce the metrics that are used in this paper to assess groundwater and baseflow drought responses (e.g. the recovery time $T_{rec}$, inter-annual variability, percentile thresholds, performance during "benchmark droughts") and how this varies spatially and temporally for Germany. Currently, the discussion in Section 2 centres on model performance for $T_{max}$ which is based on correlations and not focused on the (likely) non-linear drought responses that are being assessed here.

*The generic scenarios in the paper focus on sensitivities during drought. We agree that the model's ability to simulate the dynamics targeted in the scenarios is crucial for the reliability of the results. Hence, we will expand our reflections on the abilities and limits of the groundwater model. Specifically, in the revised manuscript we will define the required model ability for each scenario type (see Table R1) and discuss the model evaluation in these specific regards. However, we think that $T_{max}$ is still a very important evaluation metric to understand model behaviour and particularly the non-linearity of baseflow and groundwater head response: Overall $T_{max}$ for baseflow is much shorter than for groundwater directly relating to a larger dependency on intra-annual climate dynamics for baseflow and on inter-annual dynamics for groundwater heads. In Hellwig et al. (2020) it was demonstrated that these differences, which lead to the non-linearities found in our study, are appropriately captured by the model. We will state this importance of $T_{max}$ for the interpretation of the results more clearly in the revised manuscript.*

*Table R1: Required model ability and discussion of model performance for the different scenarios.*

| | Required model ability | Evaluation metric | Discussion of model performance |
|---|---|---|---|
| $S_{SHIFT}$ | Reliable propagation of inter- and intra-annual recharge dynamics into groundwater heads and baseflow | $T_{max}$ | Overall, the model depicts both differences of $T_{max}$ across the study area and the systematically shorter $T_{max}$ of baseflow compared to groundwater. However, for baseflow $T_{max}$ was notably overestimated in the North and underestimated in the South while for groundwater it was overestimated in the porous aquifers of the lowlands and underestimated in higher elevations (see Hellwig et al., 2020 for more detailed analyses). Hence, |

| | | | |
|---|---|---|---|
| | | | absolute $S_{SHIFT}$ responses may be biased in that same way. The model estimates allow most confidence in the representation of general shift-patterns across the study area. |
| $S_{EVENT}$ | Reliable model representation of benchmark drought events | Differences between observed and modelled groundwater/ baseflow drought severities | Simulations and observations show a considerable variability of groundwater drought severity for different drought years across the study area. Consistent with observations, modelled drought severities were weaker in 2003 compared to 1973 with several regions in the study area not in groundwater drought. These patterns are also consistent with state agency reports (see Hellwig et al., 2020). However, especially in the Northeast the model responds too slowly (corresponding with too long $T_{max}$, see above) leading to deviating groundwater drought severities: the drought severity of 1973 is overestimated in the model while it is underestimated for 2003. For baseflow model performance is similar: while general patterns of drought severity can be depicted, drought severities deviate most in the North (-East) (see also Figure R1). Overall, there are systematic uncertainties arising from the comparison of observational data with model outputs which might relate to some of the differences found (for a more advanced discussion on that see Hellwig et al., 2020, Section 2.3). |
| $S_{RECOV}$ | Reliable representation of severe drought + propagation of recharge forcing into groundwater | Combination of evaluation metrics of $S_{SHIFT}$ and $S_{EVENT}$ | As both general patterns of drought severities and the propagation of the forcing into groundwater are captured by the model, prerequisites for an appropriate drought termination simulation are given. Uncertainties for this scenario are – similar to the other scenarios – largest in regions of weaker model performance regarding $T_{max}$. |

[Figure]

*Figure R1: Simulated and observed anomalies averaged for summer months (JJA) of the benchmark drought years 1973 and 2003. Figure based on data taken from Hellwig et al. (2020).*

*We think with these additional remarks model results will be better interpretable to the reader while maintaining the focus of the study which are rather the different sensitivities found and not the model design and evaluation. In light of the commentary published by Gleeson et al. (2020) in the meantime, who discuss the issue of difficult-to-validate groundwater models with local observations, we suggest that we may add a more general discussion and conclusion on the issue.*

4) **Minor Comments and Technical Corrections**

**Abstract L7.** Please change to "depend on the systems' sensitivity"

**Introduction L25-28.** I would move (or remove) the two sentences starting with "Contrary to surface water, groundwater is hard to…" to L44 where you discuss the absence of observational data and use of groundwater models in more detail.

**Introduction L49.** Replace "more and more" with "increasingly"

**Introduction L55.** "Climate models (often) lack alterations in the sequencing of future wet and dry spells". This sentence needs to be supported by some references.

**Equation 1 L114.** What does the 'f' denote?

**Section 3 L164.** It is not entirely clear to me how you calculate inter-annual variability – can you clarify and provide the equation?

**Discussion.** It might be worth adding some sub-section headings to the discussion to break up the text a little for the reader.

**Discussion L267-268** "Also, the recovery time Trec from a severe drought varied accordingly (SRECOV)." I am not sure what you mean here – can you clarify?

**Supplementary Information.** Figures S1-S4 are very difficult to interpret and the figure quality is poor (i.e. they are quite blurry). Can you make the maps bigger and ensure the figures are incorporated at high resolution so that they are clear to the reader.

***Thanks for pointing out. We will correct for this in the revised version.***

***References cited in this response:***

Gleeson, T., Wagener, T., Döll, P., Zipper, S. C., West, C., Wada, Y., ... & Oshinlaja, N. (2020). HESS Opinions: Improving the evaluation of groundwater representation in continental to global scale models. *Hydrology and Earth System Sciences Discussions*, 1-39.

Hellwig, J., de Graaf, I. E. M., Weiler, M., & Stahl, K. (2020). Large-Scale Assessment of Delayed Groundwater Responses to Drought. *Water Resources Research*, *56*(2), e2019WR025441.

---

## Author Response (AR1)

We would like to thank the two anonymous Referees and Jim Freer for their feedbacks and helpful suggestions on this manuscript. Below we give point-by-point responses to all the comments (bold and italic). The new manuscript with tracked changes can be found below the responses.

**Editorial comments:**

The two core matters seem to be:

1) The tests are rather arbitrary, limited, and they are not comprehensive in nature - I also noted on the submission that they seem rather simplistic and not well justified in terms of any actual expected climatic variation - synthetic or not. I believe this needs to be resolved by improving the experimental base for the analyses, and I believe the authors intend to do this but they still need to be linked to some justification of 'climate change' else the paper should simply be about a sensitivity analyses and I am not sure that is enough for publication. So whilst the authors offer more % changes this is not enough to tie in better and more objectively with potential future scenarios and so the reasoning still has to be improved

We expanded the experimental base of our study by performing more model runs resulting in more robust and reliable findings (see answers to question #1 of reviewer 1 and question #2 of reviewer 2). Additionally, we improved the manuscript to make clear that our approach is complementary and different from the climate projections ensemble approach and starts with testing the general sensitivity of the system first. Subsequently this sensitivity can be used to assess the vulnerability of the system – not limited to but with particular relevance to expected climate change. For putting the results of the stress tests in the context of model projected climate change, of course projections must be considered in addition.

In the revised manuscript, we now consistently use 'stress test' instead of 'scenario' to avoid confusion. We better introduce, argue and explain this approach in the introduction (Section 1), eliminated all obscuring connections to climate change projections from the stress test descriptions (Section 3) and put a reflection regarding climate change in the discussion (Section 5). We think that this reorganization makes the reasoning of the manuscript now clearer and puts the insights for climate change adaptations in the right perspective.

2) That the model evaluation is improved - in the sense that some understanding of caution (or indeed not reported) for regions where the model does not do that well - which I add the authors have in some ways explored within their WRR paper. I believe I also noted this matter of model quality in my initial assessment.

We significantly expanded the manuscript regarding model evaluation specifically for the properties relevant in the stress tests and added our reflections to Section 3, 5 and the Supplement (for details see our response to comment #3 of reviewer 2).

**Referee #1:**

1) The paper describes a study where a large-scale, high-resolution MODFLOW-groundwater model of Germany has been used to assess a range of potential changes to groundwater and baseflow drought hazard based on

three change scenarios. The scenarios are: i.) a changed recharge regime with wetter winters and drier summers (SSHIFT), ii.) changes to antecedents conditions associated with three major historic episodes of drought in Germany (SEVENT), and iii.) recovery from drought (SRECOV). These scenarios were co-designed in part with the Climate and Water Initiative of southern Germany's federal states (KLIWA) (L67-86) with the aim of stress testing the sensitivity to drought of groundwater and baseflow. Although, the geographical focus of the study is Germany, the paper addresses questions relevant to a wide readership and is clearly in the scope of HESS.

The description of the model setup (Section 2) is adequate given that more details can be found in the paper by Hellwig et al. (2020) who developed the model. However, a critical assessment by the authors of the model's suitability, including method of calibration and appropriateness of it's underlying assumptions, for the current application would be helpful. The description of the scenario design and modelling approach (Section 3) is generally clear and well-reasoned. However, the scenarios appear somewhat arbitrary. In particular, the formulation of the SRECOV scenario is less convincing than the other two scenarios. To assess the maximum duration for groundwater recovery from severe drought, the lowest simulated groundwater heads are taken as an initial condition and groundwater heads are simulated using long-term average monthly recharge as input until an arbitrary recovery has been achieved. Although adequately described, the motivation and justification for the details of this scenario are not given. The results are presented well, both graphically and in their description in Section 4.

**The national-scale groundwater model is certainly limited in its local representation. Therefore, we expanded the manuscript regarding model evaluation and added our assessment specifically for the stress tests to Section 3, 5 and the Supplement (for details see our response to comment #3 of reviewer 2).**

We agree that the formulation of  $S_{RECOV}$  might appear first of all arbitrary, but it is in fact similar to wellestablished probabilistic real time forecasting practices. We selected this stress test, however, to address the stakeholders request for a better understanding of the drought termination period. To account for the local differences in the groundwater system, we adopted the idea of a 'composite map' and did not select one specific drought year but rather the lowest heads of the simulated period. We agree that the recovery threshold is arbitrary, however, additional analyses with other thresholds (40 and 50-percentile groundwater head) resulted in the same patterns (see section 4.3 of the revised manuscript). To test the influence of the recharge on  $T_{rec}$ , as suggested, we performed two additional model runs for the revised manuscript representing dry (wet) conditions during drought recovery. Accordingly, drought recovery decelerated (accelerated). However, general spatial patterns remained similar as  $T_{rec}$  is strongly related to  $T_{max}$ , which is an aquifer characteristic independent of meteorology/recharge or initial conditions (see Figure 9 c) + d) of the revised manuscript).

---

## Referee Report (RR1)

The authors have undertaken a comprehensive revision of the paper in response to comments from the Editor and two reviewers addressing all items on a point-by-point basis. Importantly, the revisions have effectively addressed the core concerns related to the relatively arbitrary nature and limited evaluation of the original experimental design. The reframing of the paper in terms of stress testing, the addition of new analyses related to recovery threshold, and additional text discussing model performance and evaluation and the implications of the results in the context of the stakeholder community have improved the paper significantly.

---

## Author Response (AR2)

We would like to thank the two anonymous Referees and Jim Freer for their positive responses to our revisions. Below we give point-by-point responses to the comments of the second round of reviews (bold and italic). The new manuscript with tracked changes can be found below the responses.

**Editorial comments:**

- 5 The reviewers are now more favourable to the paper with all your edits and clarifications and this is appreciated. However there are some questions to respond to and please can you do some work on responding to those. If this is done to satisfaction (these are more minor edits now and relate to the tone and delivery of some messages in the paper) I will review the response and changes and decide if final publication is warranted. I do however want to note I do want you to pay special attention to any aspects that suggest, without any
- 10 evidence being directly presented, that these stress tests relate to 'expected climate change". For example I see the follow quote is still in the revised manuscript (as an example). "The assumed changes of SSHIFT lie in the range of potential precipitation changes for winter and evapotranspiration changes in summer predicted by regional climate and water balance models for Germany until the end of the 21st century (Jacob et al., 2012, Herrmann et al., 2016, Paparrizos et al., 2018)" I keep on stressing this but you will have to provide direct
- 15 evidence between your scenarios and these changes along with such statements else the reader cannot judge. So please either provide an appropriate analyses to show they are 'similar' to how you have formulated your stress tests or remove such comments...

We removed the respective section and thoroughly checked and revised the manuscript to make clear that our experiments and results are not directly comparable to climate change studies based on climate model projections (see also addition in response to R2).

**Referee #2:**

20

25

The authors have carefully considered and responded to my comments. The addition of Table 2 was a nice summary of model performance for the different regions and it was worthwhile including additional stress tests.

I have a few minor revisions the authors should consider before publication (note all line numbers refer to the new version of the manuscript):

Intro L53-64 - As you now use the term 'stress test' throughout the manuscript, it would be useful to point
 out more clearly to the reader the differences between stress tests and scenarios here.

**Two sentences inserted**

2. Study area L89. Not sure what you mean by 'natural regions'? Do you mean these areas are relatively free of human influences?

**No, we simply mean major geographical regions and changed the term**

35 3. Study area L98-102. It would be useful here to give an indication of the magnitude of uncertainty from these studies. What is the range (or uncertainty) in the change of recharge?

**We added this information**

4. Discussion 5.2 L323-355. Given some of the poorer model performance noted in Table 2 - it would be interesting to add a sentence or two to the discussion where future model improvements are needed. Is the

40 anthropogenic influences the main one or are there others that you think could help improve the models ability to represent drought severity?

**For all the points discussed in section 5.2 a reduction of uncertainties could make model outputs more reliable. However, from out perspective the most relevant point is the parametrization of the groundwater model, which is limited by available hydrogeological data but drives the stress test responses. We point this out in the new manuscript.**

5. Discussion 5.2 L340-341. There needs to be a little more reflection on the stress tests you have applied and some of their limitations. I think you should add a caveat after the sentence 'The spatially different groundwater sensitivities identified in this study allow to assess the general potential of changes of groundwater and baseflow drought in a changing climate', explaining some of the limitations in your scenarios. For example, you

50 note that recharge is extremely variable with large year-to-year variations, yet you apply a constant percentage increase/decrease. How could we improve these stress tests for future studies and particularly when we are working with practitioners?

In the SSHIFT stress test we opted to concentrate on modifications of average recharge conditions, but still the large year-to-year variability is preserved in this stress test. As a side effect of the intraannual shift of

55 recharge in SSHIFT, the year-to-year variability is also changing (see Fig. 4). We agree that a systematic change of interannual variability is another potential stress that could be tested in a stress test if relevant for practitioners. Anyway, stress test design and evaluation metrics can be (and should be) adopted for specific interests of the stakeholders. We reformulated the paragraph accordingly.

[revised manuscript text omitted]

into groundwater